# Efficient Hyper-Parameter Search for LoRA via Language-aided Bayesian Optimization

## Abstract

Fine-tuning Large Language Models (LLMs) with Low-Rank Adaptation (LoRA) enables resource-efficient personalization or specialization, but it comes at the expense of additional hyperparameter tuning. Although LoRA makes fine-tuning efficient, it is highly sensitive to the choice of hyperparameters, and exhaustive hyperparameter search is still computationally very demanding. To address these challenges, we propose a framework that integrates the domain knowledge of pre-trained LLMs into Bayesian Optimization (BO) to efficiently search for LoRA hyperparameters. To leverage the informed knowledge of LLMs, we repurpose LLMs as a discrete-to-continuous mapping to link the hyperparameters and their domain knowledge with a continuous vector space, where BO is conducted. We design and control the mapping by language prompting, where we provide a domain-aware textual prompts describing the relationships among hyperparameters and their respective roles; thereby, we explicitly inject domain knowledge about LoRA into the LLM in natural language. Also, we model the residual information hard to be linguistically described in the prompt with an additional learnable token. This aids BO to sample more high-performing hyperparameters. In addition, by leveraging the observation of the strong correlation between the respective performance obtained from full and subset training datasets in LoRA training regimes, we introduce proxy training and evaluation with a data subset. This further increases the efficiency of our method. We demonstrate that our hyperparameter found with only about 30 iterations achieves more than 20% performance improvement over standard hyperparameters found from about 45,000 combinations. *Code will be released upon acceptance.*

## 1 Introduction

Large Language Models (LLMs) (Touvron et al., 2023; Team, 2024; Team et al., 2024) have been recognized as strong foundation models that can be easily adapted to diverse downstream tasks with high performance. However, fully fine-tuning LLMs for specific applications is computationally heavy. It requires updating billions of parameters, which demands substantial memory and computational resources (Brown et al., 2020; Gururangan et al., 2020). To overcome these limitations, Parameter-Efficient Fine-Tuning (PEFT) (Houlsby et al., 2019; Ding et al., 2023) methods have emerged as effective alternatives, enabling strong task adaptation at significantly reduced cost. Among these approaches, Low-Rank Adaptation (LoRA) (Hu et al., 2022) stands out as one of the most widely adopted techniques. LoRA freezes the pre-trained weights and introduces lightweight, trainable low-rank adapters, allowing models to adapt efficiently to new tasks with only a fraction of the parameters and resources required for full fine-tuning.

Despite its effectiveness, identifying an *optimal* hyperparameter configuration for LoRA remains challenging, as performance is highly sensitive to hyperparameter choices (Sengupta et al., 2024; Biderman et al., 2024; Mao et al., 2025). LoRA involves several key hyperparameters, including the rank ($r$), scaling factor ($\alpha$), batch size, learning rate, and dropout rate, which are entangled in complex ways. Consequently, performance can vary significantly depending on their combinations (Halfon et al., 2024; Sengupta et al., 2024; Mulakala et al., 2024). Therefore, systematically searching for the appropriate configuration is a critical issue. Yet, naïve exploration is infeasible: the hyperparameter search space is combinatorially large, and each evaluation is extremely costly (Valipour et al., 2022; Chavan et al., 2023; Sun et al., 2024; Meo et al., 2024; Bini et al., 2025).

This challenge motivates the use of Bayesian optimization (BO) as a principled framework for Hyperparameter Optimization (HPO). BO has proven highly effective in real-world applications where target function evaluations are expensive, such as drug discovery and materials design (Korovina et al., 2020; Ranković & Schwaller, 2023; 2025). BO relies on a surrogate model to approximate the black-box function defined by hyperparameters and their performance, and uses an acquisition function to select the next configuration by balancing exploration and exploitation. However, directly applying BO to LoRA HPO is non-trivial, since traditional BO methods require the underlying function domain to be continuous while the hyperparameters involve discretes and have no way to integrate the domain knowledge during the optimization process (Yan et al., 2025).

In this work, we propose an efficient BO-based HPO framework tailored to LoRA that incorporates domain knowledge through the LLM. Specifically, hyperparameter configurations are expressed as structured text templates describing each hyperparameter's name, value, role, and interactions. An LLM processes this template along with a learnable token and converts it into a continuous embedding, where domain knowledge is effectively encapsulated in the learnable token. The learnable token, paired with observed performance, is then used to train a surrogate model, which in turn proposes the hyperparameter candidates that maximize the acquisition function. To further improve efficiency, we introduce a proxy training evaluation that significantly reduces evaluation cost and iteration time, enabling faster and more sample-efficient optimization.

Our framework generalizes beyond LoRA to its variants, including DoRA (Liu et al., 2024a), rsLoRA (Kalajdzievski, 2023), and PiSSA (Meng et al., 2024), and is compatible across different model architectures. Experimental results demonstrate consistent performance improvements when applying our HPO framework across diverse settings. Moreover, our approach proves both more efficient and effective than existing search methods (Oliver & Wang, 2024; Tribes et al., 2024) and alternative optimization strategies (Bergstra & Bengio, 2012; Akiba et al., 2019; Li et al., 2021). Finally, by analyzing the experimental results, we observe that previously unexplored hyperparameter combinations can also deliver strong performance, offering insights for new guidelines.

In summary, our contributions are as follows:

- **The first framework combining an LLM with BO specialized for LoRA HPO.** We propose an efficient BO-based LoRA HPO framework that integrates domain knowledge into the optimization process by leveraging an LLM. This framework enables the selection of appropriate hyperparameters from a vast number of possible combinations.
- **Improving efficiency of the proposed framework.** We introduce a projection layer and a learnable token to accelerate BO process. We also introduce a proxy training evaluation protocol that significantly reduces computational cost, enabling efficient optimization.
- **Empirical validation of efficiency and generalizability.** We demonstrate the generalizability of our framework across LoRA variants and model architectures, showing consistent improvements and offering new insights into effective hyperparameter configurations.

## 2 RELATED WORK

**Low-Rank Adaptation (LoRA) and hyperparameter sensitivity in LoRA**. LoRA (Hu et al., 2022) has become one of the most widely adopted parameter-efficient fine-tuning (PEFT) methods (Houlsby et al., 2019) for Large Language Models (LLMs). By introducing a trainable low-rank adapter into a frozen pre-trained model, LoRA allows efficient task-specific adaptation without updating the full set of model parameters. Building on this idea, various LoRA variants have been proposed to improve stability, convergence, and performance. For example, DoRA (Liu et al., 2024a) decomposed each weight into a fixed magnitude and a learnable low-rank direction and rsLoRA (Kalajdzievski, 2023) rescaled LoRA updates by a factor of $\alpha/\sqrt{r}$ to improve stability. Meng et al. (2024) suggest PiSSA leveraging the principal singular vectors and values of the original weights to initialize LoRA adapters for faster convergence and performance improvement.

Although several advanced LoRA variants have been proposed, the common issue of sensitivity to hyperparameter selection remains a challenge. In particular, rank ($r$) (Zhang et al., 2024), scaling factor ($\alpha$) (Liu et al., 2025), learning rate (Jin et al., 2023), batch size (Marek et al., 2025), and dropout rate (Lin et al., 2024) identified as key factors that influence final results. This sensitivity often leads to large performance variance and complicates fair comparisons across methods.

Moreover, the "optimal" configuration frequently depends on the dataset and base model in use (Rajabzadeh et al., 2024; Yan et al., 2025). Consequently, systematic approaches for optimizing LoRA hyperparameters remain underexplored, as it is difficult to identify effective configurations while accounting for all these factors. Prior work has explored black-box optimization methods (Inouye et al., 2024; Tribes et al., 2024; Oliver & Wang, 2024; Sengupta et al., 2024) and efficient grid-search strategies (Yan et al., 2025) for LoRA hyperparameter selection. Nevertheless, these approaches commonly suffer from two limitations: (i) domain knowledge is not incorporated into the optimization process, and (ii) evaluation remains costly. Hyperparameter optimization generally requires substantial domain knowledge (Wu et al., 2019; Shawki et al., 2021; Czako et al., 2021; Bowler et al., 2022), and LoRA is no exception due to its adapter-specific properties (Halfon et al., 2024; Yan et al., 2025). To address these limitations, we propose a framework that integrates Bayesian optimization and an LLM. This framework can automatically and effectively identify suitable hyperparameters for LoRA, reducing the need for extensive manual tuning.

**Bayesian optimization for hyperparameter optimization**. Hyperparameter optimization is a critical task that significantly impacts model performance in machine learning. However, evaluating each configuration is often expensive due to the high cost of training. In this context, Bayesian optimization has emerged as a prominent method for HPO, especially in expensive evaluation settings (Snoek et al., 2012; Shahriari et al., 2015). BO uses a surrogate model and acquisition function to efficiently search for high-performing hyperparameters with fewer evaluations.

Although BO is an effective approach, its application in discrete input space such as LoRA is limited (Oh et al., 2019; Deshwal & Doppa, 2021; Chu et al., 2024). To mitigate this, several studies (Zhang et al., 2023; Ramos et al., 2023; Agarwal et al., 2025) have shown that hybrid frameworks combining LLMs with BO represent a promising direction, achieving empirical gains across diverse domains. Such approaches include using LLM agents to propose candidate hyperparameter configurations (Liu et al., 2024b), reformulating BO tasks in natural language to flexibly incorporate search spaces and constraints (Liu et al., 2024c), and enhancing surrogate models with LLM embeddings as input features (Nguyen et al., 2024). These synergies between LLMs and BO extend beyond HPO to other domains, further emphasizing their effectiveness (Ranković & Schwaller, 2023; 2025). Building on this trend, we propose the first framework that integrates BO with LLMs for LoRA HPO. We construct an embedding space tailored to LoRA HPO using an LLM with domain prompting and learnable tokens, and perform BO within this space, improving search efficiency under high-cost evaluation conditions.

## 3 METHOD

We propose a framework that combines a Large Language Model (LLM) with Bayesian optimization (BO) to discover appropriate hyperparameters for LoRA tuning in each task. We obtain continuous embeddings from the LLM and use them as inputs to the surrogate model, enabling a BO process tailored to LoRA Hyperparameter Optimization (HPO). The LLM in our framework not only encodes rich prior knowledge through large-scale pretraining, but also provides a convenient interface for injecting additional knowledge in textual form. Furthermore, to reduce cost, we introduce proxy training evaluation, which estimates the performance of a full-dataset model using a model trained on a subset of the data. With these components, our framework improves not only the sample efficiency of BO, but also the computational efficiency of LoRA hyperparameter optimization as a whole. Section 3.1 introduces the preliminaries of BO, Sec. 3.2 presents the proposed framework and its components, and Sec. 3.3 details our proxy training evaluation.

### 3.1 PRELIMINARY: BAYESIAN OPTIMIZATION

BO is an efficient approach for optimizing black-box functions, particularly when the evaluation cost is expensive. The goal of BO is to find the optimal input $\mathbf{x}^*$ from a candidate pool $\mathcal{X}$ that maximizes a black-box function $f$. The objective of BO can be formulated as follows:

$$\mathbf{x}^* = \arg \max_{\mathbf{x} \in \mathcal{X}} f(\mathbf{x}). \tag{1}$$

Since $f$ is hard to estimate, *surrogate model* $\hat{f}$ is used to approximate $f$. A common choice for the *surrogate model* is a Gaussian Process (GP), which can be expressed as: $\hat{f} \sim \mathcal{GP}(\mu, k_\omega)$, where $\mu$

is mean function and $k_\omega$ denotes the kernel function with hyperparameter $\omega$. For example, in the case of the Matérn 5/2 kernel, $\omega$ includes trainable hyperparameters,

$$k_\omega(\mathbf{x}, \mathbf{x}') = \sigma^2 \left(1 + \frac{\sqrt{5}d}{\ell} + \frac{5d^2}{3\ell^2}\right) \exp\left(-\frac{\sqrt{5}d}{\ell}\right), \tag{2}$$

where $\ell$ denotes the lengthscale, $\sigma^2$ denotes covariance and $d = \|\mathbf{x} - \mathbf{x}'\|_2$. Given $n$ observed points set $\mathcal{D}_n = \{(x_i, y_i)\}_{i=1}^n$, the surrogate model is tuned on observed points, and an acquisition function $\alpha$ is used to determine the next evaluation point $\tilde{\mathbf{x}}$ based on the posterior from the *surrogate model*:

$$\tilde{\mathbf{x}} = \arg\max_{\mathbf{x} \in \mathcal{X}} \alpha(\mathbf{x}|\hat{f}, \mathcal{D}). \tag{3}$$

## 3.2 PROPOSED FRAMEWORK

**Overview**. Our framework performs 4 steps in each iteration: (1) Proxy training evaluation ($Proxy$), which fine-tunes LoRA on a subset of the dataset and measures its performance; (2) Embedding extraction using the LLM; (3) Surrogate model update; and (4) Next evaluation point suggestion. For example, in the $n$-th iteration, a hyperparameter configuration $x_n$ is selected, and its benchmark performance $y_n$ is obtained through proxy training evaluation. The configuration $x_n$ is then converted into a structured template $t_n$ via domain-aware prompting. This template, together with the learnable token $\psi$, is passed into the LLM $\phi$ and projection layer $P(\cdot; \theta)$ to produce an embedding: $\mathbf{z}_n = P(\phi(t_n, \psi); \theta)$. The surrogate model parameterized $\omega$ is updated by maximizing the *marginal log-likelihood* using embedding $\mathbf{z}_n$ paired with the observed target $y_n$, jointly updating all trainable parameters $\omega, \theta$, and $\psi$. Finally, the next evaluation point is selected by generating embeddings for every hyperparameter configuration $x_j$ in the candidate pool $\mathcal{X}_{cand}$ and evaluating the acquisition function $\alpha$. Algorithm 1 describes the entire procedure in pseudo-code.

---

**Algorithm 1** Pseudo code for our framework

---

**Require:** Candidate pool $\mathcal{X}_{\text{cand}}$, observed dataset $\mathcal{D}_n = \{(x_i, y_i)\}_{i=1}^n$, budget $N$,
    parameters $\omega$ (GP), $\theta$ (Projection layer), $\psi$ (Learnable token),
    LLM $\phi$, acquisition function $\alpha$, feature extractor $g(\cdot; \theta, \psi)$
**Initialize:** parameters $\omega, \theta, \psi$; $\mathcal{D}_0 \leftarrow \emptyset$; Choose initial candidate $x_1 \in \mathcal{X}_{cand}$
  1: **for** $n = 1$ to $N$ **do**
  2:     $y_n \leftarrow Proxy(x_n)$                         $\triangleright Proxy$ means **Proxy training evaluation**
  3:     $\mathcal{D}_n \leftarrow \mathcal{D}_{n-1} \cup \{(x_n, y_n)\}$
  4:     Remove $x_n$ from $\mathcal{X}_{\text{cand}}$
  5:     **while** not convergence **do**                      $\triangleright$ Surrogate model update
  6:         **for all** $x_i \in \mathcal{D}_n$ **do**
  7:             $t_i \leftarrow Template(x_i)$       $\triangleright Template$ means **Domain-aware prompting**
  8:             $\mathbf{z}_i \leftarrow g(\mathbf{x}_i; \theta, \psi) = P(\phi(t_i, \psi); \theta)$
  9:         **end for**
10:         Compute *marginal log-likelihood* $\log p(\mathbf{y}|\mathbf{Z}, \omega, \theta, \psi)$
11:         Update $\omega, \theta, \psi$
12:     **end while**
13:     **for all** $x_j \in \mathcal{X}_{\text{cand}}$ **do**                   $\triangleright$ Bayesian optimization
14:         $t_j \leftarrow Template(x_j)$
15:         $\mathbf{z}_j \leftarrow g(\mathbf{x}_j; \theta, \psi) = P(\phi(t_j, \psi); \theta)$
16:         Compute $\alpha(\mathbf{z}_j; \omega, \theta, \psi)$
17:     **end for**
18:     $j\prime = \arg\max_j \alpha(\mathbf{z}_j; \omega, \theta, \psi)$
19:     Suggest next evaluation point $x_{n+1} \leftarrow x_{j\prime}$
20: **end for**
21: $(x^*, y^*) \leftarrow \arg\max_{(x,y) \in \mathcal{D}} y$
22: **return** $x^*$

---

**Domain-aware prompting**. We employ domain-aware prompting to explicitly incorporate domain knowledge about LoRA hyperparameters into the optimization process. A straightforward text template can be written as $t = \{\text{name}, \text{value}\}$ (Table A10a). However, this simple format fails to

capture the roles or relationships between hyperparameters. Prior studies have highlighted practical know-how and manual tuning guidelines for hyperparameter tuning (Mohammed & Kora, 2025; He, 2024; Diehl, 2024; unsloth, 2025). For example, Hu et al. (2022) suggest that the scaling factor ($\alpha$) in LoRA behaves similarly to adjusting the learning rate. To better reflect existing know-how and guidelines, we design a structured text template $t = \{\text{explanation}, \text{name}, \text{value}\}$ (Table A10b). The explanations emphasize relationship between hyperparameters (*e.g.* how rank and alpha are commonly set) as well as the training dynamics that arise when their magnitudes vary. This approach goes beyond simple LLM embeddings, enabling the direct insertion of structured, prompt-level information into the embedding space.

**Learnable token and projection layer**. Calibrating the embedding space extracted from the LLM during feature extraction can enhance BO performance compared to using fixed embeddings (Kristiadi et al., 2024; Ranković & Schwaller, 2025). Motivated by this insight, we introduce a learnable token $\psi$ along with a projection layer $P(\cdot; \theta)$, parameterized by $\theta$, to transform embeddings into a space better suited for BO. We append the learnable token to the domain-aware text template $t$ and feed both into the LLM to extract embeddings, allowing the token to compactly capture LoRA hyperparameter's knowledge. These embeddings are then passed through the projection layer, producing representations tailored for BO. Throughout this process, the pre-trained LLM remains frozen, while $\psi$ and $\theta$ are learnable. The final embedding is obtained via pooling the embedding at last token's position, resulting in the final feature: $\mathbf{z} = P(\phi(t, \psi); \theta)$. As a result, the embedding not only explicitly reflects the explanations encoded in the prompt, however also implicitly internalizes LoRA-specific domain knowledge. This improves representational power and enhances BO efficiency with minimal additional parameters.

**Bayesian optimization with LLM**. BO typically employs Gaussian Processes (GPs) as surrogate models, which are effective for modeling distributions over continuous spaces (Beckers, 2021). However, when dealing with complex input spaces that require understanding the relationships among variables, it becomes crucial to use representations capable of capturing such structure (Lee et al., 2025). This is particularly true for the LoRA hyperparameter space, which is inherently discrete and requires domain knowledge. To address this challenge, we integrate an LLM with a learnable token and a projection layer, which inject domain knowledge about LoRA HPO into when extracting embeddings: $\mathbf{z} = g(\mathbf{x}; \theta, \psi) = P(\phi(t, \psi); \theta)$. Therefore, we employ LLM-based deep kernel learning to combine the prior knowledge encoded in the LLM with these trainable neural architecture for the GP, thereby transforming standard GP regression into deep kernel learning:

$$k(\mathbf{x}, \mathbf{x}'|\omega) \rightarrow k(g(\mathbf{x}; \theta, \psi), g(\mathbf{x}'; \theta, \psi)|\omega, \theta, \psi). \tag{4}$$

We jointly optimize all trainable parameters, $\Phi = \{\omega, \theta, \psi\}$, where $\omega, \theta$, and $\psi$ denote the GP kernel, projection layer, and learnable token parameters, respectively. These are optimized by maximizing the marginal log-likelihood:

$$\mathcal{L}(\Phi) = \log p(\mathbf{y}|\mathbf{X}, \Phi) = -\frac{1}{2}\{(\mathbf{y} - \mu\mathbf{1})^\top \mathbf{K}_\Phi^{-1}(\mathbf{y} - \mu\mathbf{1}) + \log|\mathbf{K}_\Phi| + n \log 2\pi\}, \tag{5}$$

$$\Phi^* = \arg\max_\Phi \mathcal{L}(\Phi), \tag{6}$$

where $\mathbf{K}_\Phi$ denotes covariance matrix determined from the covariance kernel of the GP, $\mathbf{X} = \{\mathbf{x}_1, \mathbf{x}_2, ..., \mathbf{x}_n\}$ and $\mathbf{y} = \{y_1, y_2, ..., y_n\}$.

### 3.3 PROXY TRAINING EVALUATION

Previous studies (Klein et al., 2017; Oliver & Wang, 2024) have shown that it is not always necessary to train on the entire dataset at every optimization step because training performance on subset datasets strongly correlates with that of full training. Building on these insights, we introduce a proxy training evaluation strategy to reduce fine-tuning time cost. Specifically, instead of training on the full dataset, we fine-tune the model on a randomly selected subset and measure performance on this smaller training run as a proxy for the true performance. Despite its simplicity, this approach exhibits strong correlation with the true performance, and we find that using only 10% of the data can be sufficient. As a result, we reduce the overall time cost by up to 10x, enabling more optimization iterations within the same computational budget. We further consider data selection strategies such as Liu et al. (2024d), but we observe that our simple random subset achieves comparably high correlation with full-data results relative to these strategies.

## 4 EXPERIMENTS

### 4.1 EXPERIMENTAL SETTING

**LoRA hyperparameters and setup**. We define the candidate pool of hyperparameters as shown in Table 1. Specifically, we optimize five hyperparameters: rank ($r$), scaling factor ($\alpha$), learning rate, dropout rate, and batch size—resulting in a search space of more than 45,000 configurations. To validate our proposed framework, we conduct experiments across multiple LoRA variants, including rsLoRA (Kalajdzievski, 2023), DoRA (Liu et al., 2024a), and PiSSA (Meng et al., 2024).

**Tasks**. Following prior work (Meng et al., 2024), we fine-tune models on the Meta-MathQA dataset (Yu et al., 2023) and evaluate performance on GSM8k (Cobbe et al., 2021) and MATH (Hendrycks et al., 2021), and report Accuracy (%). To test generalization beyond mathematical reasoning, we extend experiments to code generation, fine-tuning on the CodeFeedback dataset (Zheng et al., 2024) and evaluating on HumanEval (Chen et al., 2021) and MBPP (Austin et al., 2021), reporting Pass@1 which is the probability that the first generated solution solves the task. Each training dataset contains 100K samples, with a 10K subset used for proxy training evaluation.

Table 1: **Hyperparameter search range.** We set the hyperparameter search ranges based on prior work (Meng et al., 2024; Wang et al., 2024; Inouye et al., 2024; Diehl, 2024; Yan et al., 2025; unsloth, 2025), resulting in a search space of over 45,000 configurations.

| Hyperparameters | Search Range | Count |
|---|---|---|
| Rank ($r$) | $1 \sim 256$ ($2^n$) | 9 |
| Scaling Factor ($\alpha$) | $\frac{r}{2} \sim 128r$ ($2^n r$) | 9 |
| Batch Size | $2 \sim 256$ ($2^n$) | 8 |
| Learning Rate | 1e-6 $\sim$ 5e-3 | 10 |
| Dropout Rate | $0.0 \sim 0.3$ ($0.5 \times n$) | 7 |

**Baselines**. We benchmark our framework against several HPO methods: random search (Bergstra & Bengio, 2012), Optuna (Akiba et al., 2019), standard BO (Oliver & Wang, 2024), latent BO (LBO) (Li et al., 2021), and NOMAD (Tribes et al., 2024). Details are reported in Appendix B.

### 4.2 EXPERIMENTAL RESULTS

**Hyperparameter optimization for LoRA variants and various models**. Based on previous findings that tasks and architectures demand distinct hyperparameters (Sengupta et al., 2024; He, 2024; Mohammed & Kora, 2025), we evaluate our framework across diverse LoRA variants and models. Table 2 shows that adapting our HPO framework enables effective hyperparameter search for each LoRA variant, consistently improving performance compared to the originally reported results. Surprisingly, our framework achieves up to 21.46% accuracy improvement, emphasizing the importance of hyperparameter selection. These results suggest that there is significant room for improvement in existing LoRA variants through systematic hyperparameter search. Similarly, Table 3 demonstrates that our approach can identify appropriate model-specific hyperparameters. Across different backbone Large Language Models (LLMs), adapting our framework consistently achieves substantial improvements, highlighting its practical utility for fine-tuning newly introduced models.

Table 2: **Results of applying our framework to LoRA variants.** We set the hyperparameter configurations suggested by each work, where they were dedicatedly tuned (Kalajdzievski, 2023; Liu et al., 2024a; Meng et al., 2024; Wang et al., 2024). Using our method, we observe consistent performance improvements across all variants.

| Strategy | Ours | Accuracy (%) | | Pass@1 | |
|---|---|---|---|---|---|
| | | **GSM8K** | **MATH** | **HumanEval** | **MBPP** |
| LoRA (Hu et al., 2022) | ✗ | 41.47 | 5.24 | 16.31 | 35.47 |
| | ✓ | 62.93 (+21.46) | 12.88 (+7.64) | 30.49 (+14.18) | 42.59 (+7.12) |
| rsLoRA (Kalajdzievski, 2023) | ✗ | 41.16 | 5.46 | 16.46 | 35.72 |
| | ✓ | 58.15 (+16.99) | 10.76 (+5.3) | 29.87 (+13.41) | 42.06 (+6.34) |
| DoRA (Liu et al., 2024a) | ✗ | 40.11 | 5.36 | 17.07 | 36.51 |
| | ✓ | 57.01 (+16.9) | 10.78 (+5.42) | 30.58 (+13.51) | 42.33 (+5.82) |
| PiSSA (Meng et al., 2024) | ✗ | 52.46 | 7.34 | 22.56 | 40.48 |
| | ✓ | 60.88 (+8.42) | 12.06 (+4.72) | 31.71 (+9.15) | 41.53 (+1.05) |

Table 3: **Results of applying our framework across diverse models.** We compare against the hyperparameter settings suggested by PiSSA (Meng et al., 2024), where they were dedicatedly tuned. The experiments demonstrate that our method is effective across a wide range of models.

| Model | Ours | Accuracy (%) | | Pass@1 | |
|---|---|---|---|---|---|
| | | GSM8K | MATH | HumanEval | MBPP |
| LLaMA2-7B (Touvron et al., 2023) | ✗ | 41.47 | 5.24 | 16.31 | 35.47 |
| | ✓ | 62.93 (+21.46) | 12.88 (+7.64) | 30.49 (+14.18) | 42.59 (+7.12) |
| Mistral-7B-v0.1 (Jiang et al., 2023) | ✗ | 69.90 | 19.96 | 45.73 | 61.90 |
| | ✓ | 74.07 (+4.17) | 23.46 (+3.5) | 54.27 (+8.54) | 65.08 (+3.18) |
| Gemma-7B (Team et al., 2024) | ✗ | 75.51 | 29.44 | 49.39 | 63.23 |
| | ✓ | 78.77 (+3.26) | 30.24 (+0.8) | 53.05 (+3.66) | 67.46 (+4.23) |

Table 4: **Comparison against existing HPO methods.** Our method outperforms existing HPO approaches under the same optimization budget.

| Search Method | Accuracy (%) | | Pass@1 | |
|---|---|---|---|---|
| | GSM8K | MATH | HumanEval | MBPP |
| Random | 59.14 | 10.51 | 23.17 | 36.77 |
| Optuna (TPE) | 37.38 | 4.74 | 27.44 | 38.62 |
| BO | 57.32 | 11.42 | 20.12 | 35.19 |
| LBO | 59.51 | 11.88 | 26.83 | 37.83 |
| Ours | **62.93** | **12.88** | **30.49** | **42.59** |

Table 5: **Comparison against existing LoRA HPO method.** We compare our approach with Tribes et al. (2024), which applies the NOMAD algorithm specifically for LoRA hyperparameter tuning. Our method is both more time-efficient and more effective, achieving superior performance by a significant margin. Note that H denotes hours.

| Method | Time | GSM8K | MATH | HumanEval | MBPP |
|---|---|---|---|---|---|
| Tribes et al. (2024) | 180 H | 52.16 | 9.12 | 24.39 | 37.30 |
| Ours | 24 H | 62.93 | 12.88 | 30.49 | 42.59 |

Overall, our framework can be applied as a plug-and-play module that adapts to the LoRA variants and models, giving consistent gains without manual hyperparameter tuning.

We further analyze the hyperparameter combinations discovered in our experiments. As shown in Tables A1 and A2, smaller batch sizes are often preferred, consistent with prior findings (Marek et al., 2025). In addition, applying dropout often leads to better performance. Interestingly, we sometimes observe strong performance when the scaling factor ($\alpha$) is 16 or even 32 times larger than the rank. This observation has not been reported in prior studies, where $\alpha$ was set to twice the rank according to existing guidelines (Diehl, 2024; unsloth, 2025), or determined based on a rank or fixed $\alpha$ (Kalajdzievski, 2023; Sun et al., 2024; Liu et al., 2025). This suggest that there may exist settings beyond the commonly chosen rank and $\alpha$ values that can further improve performance, thereby hinting at the possibility of proposing a new guideline.

**Comparison with various HPO methods**. We evaluate the effectiveness of our framework, against widely adopted baselines for HPO. Table 4 summarizes the results of applying each method under the same optimization budget. The results demonstrate that our approach identifies more suitable hyperparameters within a constrained budget. Notably, our method discovers better configurations than other BO-based methods, indicating that leveraging LLMs to provide domain knowledge about the search space can improve both search efficiency and effectiveness. We also compare our framework with Tribes et al. (2024), a dedicated approach for LoRA HPO that employs validation loss with the NOMAD algorithm for hyperparameter estimation. As shown in Table 5, our method finds more appropriate configurations in a shorter time. These experiments support that the combination of LLM and BO leads to improvement in both accuracy and efficiency.

**Ablation studies**. We conduct a series of experiments to evaluate the effect of each proposed component. Our framework incorporates domain knowledge into the optimization process through three components: domain-aware prompting for explicit knowledge injection, and a projection layer with a learnable token for implicitly encoding domain knowledge. As shown in Table 6, adding each component consistently helps BO to discover better-performing hyperparameter set-

Table 6: **Ablation results.** We validate each of our proposed components and find that all contribute effectively to LoRA HPO.

| Projection Layer | Domain-aware Prompting | Learnable Token | GSM8K | MATH |
|---|---|---|---|---|
| ✗ | ✗ | ✗ | 47.76 | 8.72 |
| ✓ | ✗ | ✗ | 53.98 | 9.16 |
| ✓ | ✓ | ✗ | 61.41 | 12.46 |
| ✓ | ✓ | ✓ | **62.93** | **12.88** |

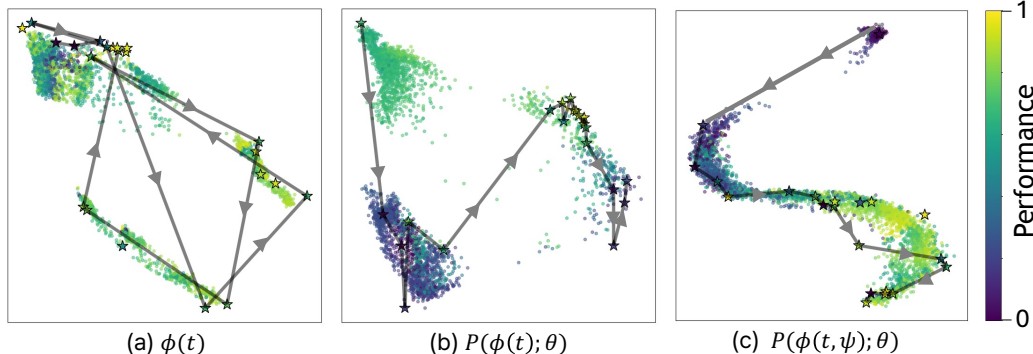

|     (a) $\phi(t)$     |     (b) $P(\phi(t); \theta)$     |     (c) $P(\phi(t, \psi); \theta)$     |

Figure 1: **Qualitative analysis of embedding space evolution using our components.** We illustrate how the embedding space evolves with our proposed components: (a) shows the embedding space from a frozen LLM $\phi$; (b) shows the space when a projection layer $P(\cdot; \theta)$ is added to the frozen LLM; and (c) shows the space when both the projection layer and the learnable token $\psi$ are employed. The trajectories in each figure indicate optimization paths across steps, shown in arrow sequence. These results suggest that incorporating the projection layer and learnable token produces a smoother, more structured embedding space suited for BO, thereby enabling efficient optimization.

tings, demonstrating the effectiveness of our framework. Notably, domain-aware prompting plays a crucial role in performance improvement, emphasizing the importance of explicitly injecting domain knowledge at the prompt level. We further analyze the differences in the optimization process introduced by each component. Without any components, BO tends to keep the learning rate nearly fixed, resulting in insufficient exploration of the search space. In contrast, BO with all components explores broadly across the hyperparameter candidate pool. These findings show that our framework enables effective exploration of diverse hyperparameters even with a small number of iterations, allowing BO to operate over a much broader search space.

**Qualitative analysis of the effect of our framework**. We visualize the embedding $z$ of hyperparameter configurations to illustrate the effect of adding each component of our framework, as shown in Fig. 1. The figure compares three settings: (a) frozen LLM embeddings, (b) embeddings after applying the projection layer, and (c) embeddings with both the projection layer and a learnable token. With frozen LLM embeddings, high- and low-performing hyperparameters configurations remain entangled, leading to an unstable search process. This results indicates that the embedding space does not effectively separate hyperparameter combinations and may hinder the balance between exploration and exploitation. Introducing a projection layer begins to separate the embeddings, revealing clearer structures that distinguish the performance levels. When we additionally incorporate a learnable token, the embeddings exhibit directional organization aligned with performance, enabling more reliable surrogate fitting and a better-organized space overall. Furthermore, we analyze the trajectories of the BO process across different settings and find that optimization using only a frozen LLM proceeds without a clear direction. In contrast, when components for embedding calibration are included, the BO process consistently moves toward the high-performing region. These observations suggest that calibration with a projection layer and a learnable token makes the BO landscape more discriminative and smooth compared to using fixed embeddings, thereby improving search efficiency and final performance.

**Validation of proxy training evaluation**. To reduce the time cost of fine-tuning during HPO, we introduce proxy training evaluation in Sec. 3.3. Using the proposed proxy training evaluation, we estimate the performance by training on a subset of the dataset, treating it as a proxy for full-data performance. To further investigate this, we examine the Pearson correlation between the training performance of subset datasets at various sampling ratios and that of the full dataset. Additionally, we compare with the existing data sampling method,

Table 7: **Correlation between the performance trained on a subset and on the full dataset.** Proxy training evaluation shows comparable correlation to full dataset accuracy, at both random sampling and TSDS (Liu et al., 2024d).

| Sampling Method | MATH Reasoning | Code Generation |
|---|---|---|
| Random (1%) | 0.7031 | 0.7429 |
| Random (5%) | 0.8360 | 0.9282 |
| Random (10%) | 0.8713 | **0.9427** |
| TSDS (10%) by Test dataset | **0.8754** | 0.9290 |
| TSDS (10%) by Train dataset | 0.8649 | 0.9278 |

Table 8: **Performance differences across model sizes.** We apply the hyperparameters discovered for each model size of Qwen2.5 to fine-tune all model sizes. "Model" denotes the model being fine-tuned, while "Settings" indicate the size of the model from which the hyperparameters were obtained. The results show that variations in model size do not significantly affect the discovery of effective hyperparameter settings.

| Model | Settings | Accuracy (%) | | Pass@1 | |
|---|---|---|---|---|---|
| | | GSM8K | MATH | HumanEval | MBPP |
| Qwen2.5-3B (Qwen et al., 2025) | 3B | 79.53 | 43.18 | 70.12 | 77.78 |
| | 7B | 78.54 | 42.40 | 68.29 | 75.13 |
| | 14B | 78.47 | 42.94 | 67.07 | 77.25 |
| Qwen2.5-7B (Qwen et al., 2025) | 3B | 84.08 | 48.90 | 81.09 | 78.84 |
| | 7B | 83.93 | 48.08 | 79.88 | 78.57 |
| | 14B | 83.09 | 48.58 | 81.71 | 78.31 |
| Qwen2.5-14B (Qwen et al., 2025) | 3B | 87.41 | 51.68 | 82.32 | 82.80 |
| | 7B | 86.81 | 50.62 | 79.88 | 81.22 |
| | 14B | 87.34 | 51.50 | 81.71 | 82.01 |

TSDS (Liu et al., 2024d), setting the target distribution to the test dataset or to the training dataset. Table 7 demonstrates that proxy training evaluation with a randomly selected 10% subset provides a sufficiently accurate approximation of full dataset performance. These results indicate that our proxy training evaluation serves as an effective and reliable indicator of model performance on the full dataset. Moreover, the correlation obtained from the 10% random subset is comparable to that of TSDS (Liu et al., 2024d) and even achieves the highest correlation in the code generation task. Based on these findings, we adopt 10% random sampling to construct the subset dataset.

**Effect of model size on LoRA HPO**. We investigate the effect of model size on finding suitable LoRA hyperparameters. Specifically, we apply our framework to Qwen2.5 models with 3B, 7B, and 14B parameters, identifying the best hyperparameters for each model size. We then use these configurations to fine-tune models across all sizes. As shown in Table 8, hyperparameter configurations discovered on one model size generally remain effective for other sizes. These results suggest that variations in model size do not significantly affect the discovery of effective hyperparameter settings. In contrast, we find differences between architecture: compared to LLaMA2, Qwen2.5 models tend to prefer smaller ranks and larger batch sizes (see Tables A1 and A2). Moreover, as shown in Table A8, when we cross-apply the configurations found with the Qwen2.5 model to LLaMA2 model and vice versa, we observe substantially larger performance degradation than when each configuration is used within the same model series. These observations indicate that hyperparameters are influenced more by model architecture than scale, which is consistent with findings from prior work (Yan et al., 2025). Since the configurations are largely transferable across scales within the same model series, this implies that tuning costs can be reduced by applying configurations found on smaller models to larger ones, rather than running the framework directly on larger models.

## 5 CONCLUSION

We propose a framework that combines Large Language Models (LLMs) with Bayesian optimization (BO) for LoRA Hyperparameter Optimization (HPO). Domain knowledge about LoRA is explicitly injected into the BO process via domain-aware prompting, while a learnable token and a projection layer transform LLM embeddings into a space better suited for optimization. To further reduce cost, we employ empirically validated proxy training evaluation, which estimates fine-tuning performance using a subset of the training data. As a result, our framework identifies appropriate hyperparameter configurations from a large candidate pool with significantly reduced optimization time. It functions as a plug-and-play module, achieving consistent performance improvements across LoRA variants, model architectures, and model scales. Comparisons with existing HPO methods validate its effectiveness both in terms of cost and performance. Beyond LoRA, we believe this framework can serve as a practical baseline for broader HPO in diverse fine-tuning strategies.

## 6 REPRODUCIBILITY STATEMENT

We provide comprehensive details of the experimental setup in the Appendix B, D, including full specifications of the hyperparameters configuration searched, and training procedures, and the prompt used for LLM feature extraction to ensure clarity and transparency. These materials are intended to make it straightforward for others to replicate our experiments and verify the reported results. In addition, we will make the complete codebase publicly available upon acceptance.

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

APPENDIX

## A   OPTIMIZING $\omega, \theta, \psi$ THROUGH MARGINAL LOG-LIKELIHOOD

This part is inspired by Wilson et al. (2016) and Ranković & Schwaller (2025), from which we partially adopt several equations. We formulate *marginal log-likelihood* as follows:

$$\mathcal{L}(\Phi) = \log p(\mathbf{y}|\mathbf{X}, \Phi) = -\frac{1}{2}\{(\mathbf{y} - \mu\mathbf{1})^\top \mathbf{K}_\Phi^{-1}(\mathbf{y} - \mu\mathbf{1}) + \log|\mathbf{K}_\Phi| + n\log 2\pi\}, \quad (7)$$

where $\mathbf{X} = \{\mathbf{x}_1, \mathbf{x}_2, ..., \mathbf{x}_n\}$ and $\mathbf{y} = \{y_1, y_2, ..., y_n\}$.

To maximize *marginal log-likelihood*, gradient-based optimization is used to optimize kernel hyperparameter $\omega$, weight of projection layer $\theta$, and learnable token $\psi$. We define the parameter set $\Phi = \{\omega, \theta, \psi\}$. The gradient of the *marginal log-likelihood* can be computed by applying the chain rule with respect to each parameter, resulting in the following decomposition:

$$\frac{\partial \mathcal{L}}{\partial \boldsymbol{\omega}} = \frac{\partial \mathcal{L}}{\partial K_\Phi}\frac{\partial K_\Phi}{\partial \boldsymbol{\omega}}, \quad \frac{\partial \mathcal{L}}{\partial \theta} = \frac{\partial \mathcal{L}}{\partial K_\Phi}\frac{\partial K_\Phi}{\partial g(\mathbf{x};\theta,\psi)}\frac{\partial g(\mathbf{x};\theta,\psi)}{\partial \theta}, \quad \frac{\partial \mathcal{L}}{\partial \psi} = \frac{\partial \mathcal{L}}{\partial K_\Phi}\frac{\partial K_\Phi}{\partial g(\mathbf{x};\theta,\psi)}\frac{\partial g(\mathbf{x};\theta,\psi)}{\partial \psi},$$
$$(8)$$

$$\frac{\partial \mathcal{L}}{\partial K_\Phi} = \frac{1}{2}K_\Phi^{-1}(\mathbf{y} - \mu\mathbf{1})(\mathbf{y} - \mu\mathbf{1})^\top K_\Phi^{-1} - \frac{1}{2}K_\Phi^{-1}, \quad (9)$$

where $\frac{\partial K_\Phi}{\partial \boldsymbol{\omega}}$ are the derivatives of the kernel with respect to the kernel hyperparameters, $\frac{\partial K_\Phi}{\partial g(\mathbf{x};\theta,\psi)}$ means the implicit derivatives of the kernel with respect to the $g$. $\frac{\partial g(\mathbf{x};\theta,\psi)}{\partial \theta}$ are the derivatives of the projection layer parameters via backpropagation and $\frac{\partial g(\mathbf{x};\theta,\psi)}{\partial \psi}$ are the derivatives of the learnable token parameters via backpropagation. Finally, we can compute the gradient of *marginal log-likelihood* by applying the chain rule.

## B   DETAILS OF THE EXPERIMENTAL SETTING

**Implementation details**.   Motivated by Ranković & Schwaller (2025), we use Qwen2-7B as the LLM in our framework to extract embeddings for BO, applying last-token pooling. The embedding dimension is set to 3584. We define the projection layer as follows:

$$P(\mathbf{x};\theta) = \text{ELU}(\text{Dropout}(W\mathbf{x} + b)). \quad (10)$$

We also utilize Matérn-5/2 kernel and Expected Improvement (EI) as acquisition function based on previous studies (Ranković et al., 2024; Ranković & Schwaller, 2025). The backbone LLM used for experiments on LoRA variants is LLaMA2-7B. For cases where our method is not applied, we follow the settings reported in prior work (Meng et al., 2024; Wang et al., 2024). We set the model's sequence length to 1024 and use a warmup ratio of 0.03 with a cosine learning rate scheduler. To reduce computational cost, all models are trained for a single epoch.

**Details on competing methods**.   We conduct our experiments on several HPO methods, random search (Bergstra & Bengio, 2012), Optuna (Akiba et al., 2019), standard BO (Oliver & Wang, 2024), latent BO (LBO) (Li et al., 2021), and NOMAD (Tribes et al., 2024). To ensure fairness, all methods are constrained to 30 optimization iterations. For each method, the top-1 result is obtained by selecting the best hyperparameter configuration on the training subset. We use the BoTorch library (Balandat et al., 2020) for BO and LBO implementations, conducting hyperparameter search with the same design space as ours. For both BO and LBO, each hyperparameter configuration is represented as a 5-dimensional vector and fed into the baselines. For LBO, we adapt the feature extractor proposed by Lee et al. (2025), which consists of two repeated blocks of linear and ReLU layers with a hidden dimension of 64. For Optuna, we use the default TPE setting with categorical hyperparameter candidates. For the method of Tribes et al. (2024), we run the NOMAD algorithm

Table A1: **Hyperparameters for math reasoning tasks.** We present the hyperparameter configuration used to train MetaMathQA for evaluating on the GSM8K and MATH datasets.

| Models | Strategy | Ours | Hyperparameter | | | | |
| --- | --- | --- | --- | --- | --- | --- | --- |
| | | | Rank($r$) | Scaling Factor($\alpha$) | Dropout | Batch Size | Learning Rate |
| LLaMA2-7B | LoRA | ✗ | 8 | 16 | 0.0 | 32 | 2e-05 |
| | | ✓ | 256 | 8192 | 0.0 | 4 | 5e-06 |
| | rsLoRA | ✗ | 8 | 16 | 0.0 | 32 | 2e-05 |
| | | ✓ | 128 | 1024 | 0.05 | 64 | 5e-05 |
| | DoRA | ✗ | 8 | 16 | 0.0 | 32 | 2e-05 |
| | | ✓ | 16 | 16 | 0.3 | 16 | 5e-04 |
| | PiSSA | ✗ | 128 | 128 | 0.0 | 128 | 2e-05 |
| | | ✓ | 256 | 4096 | 0.0 | 4 | 5e-06 |
| LLaMA2-13B | LoRA | ✗ | 8 | 16 | 0.0 | 32 | 2e-05 |
| | | ✓ | 32 | 512 | 0.0 | 2 | 5e-05 |
| Mistral-7B-v0.1 | LoRA | ✗ | 128 | 128 | 0.0 | 128 | 2e-05 |
| | | ✓ | 128 | 128 | 0.1 | 4 | 3e-05 |
| Gemma-7B | LoRA | ✗ | 128 | 128 | 0.0 | 128 | 2e-05 |
| | | ✓ | 64 | 256 | 0.0 | 2 | 5e-06 |
| Qwen2.5-3B | LoRA | ✓ | 1 | 4 | 0.25 | 32 | 5e-05 |
| Qwen2.5-7B | LoRA | ✓ | 32 | 64 | 0.25 | 16 | 5e-05 |
| Qwen2.5-14B | LoRA | ✓ | 1 | 4 | 0.25 | 32 | 2e-05 |

Table A2: **Hyperparameters for code generation tasks.** We present the hyperparameter configuration used to train CodeFeedback for evaluating on the HumanEval and MBPP datasets.

| Models | Strategy | Ours | Hyperparameter | | | | |
| --- | --- | --- | --- | --- | --- | --- | --- |
| | | | Rank($r$) | Scaling Factor($\alpha$) | Dropout | Batch Size | Learning Rate |
| LLaMA2-7B | LoRA | ✗ | 8 | 16 | 0.0 | 32 | 2e-05 |
| | | ✓ | 256 | 128 | 0.0 | 4 | 5e-05 |
| | rsLoRA | ✗ | 8 | 16 | 0.0 | 32 | 2e-05 |
| | | ✓ | 256 | 128 | 0.25 | 4 | 5e-05 |
| | DoRA | ✗ | 8 | 16 | 0.0 | 32 | 2e-05 |
| | | ✓ | 128 | 256 | 0.15 | 2 | 3e-05 |
| | PiSSA | ✗ | 128 | 128 | 0.0 | 128 | 2e-05 |
| | | ✓ | 32 | 1024 | 0.0 | 2 | 3e-05 |
| LLaMA2-13B | LoRA | ✗ | 8 | 16 | 0.0 | 32 | 2e-05 |
| | | ✓ | 256 | 128 | 0.25 | 2 | 1e-04 |
| Mistral-7B-v0.1 | LoRA | ✗ | 128 | 128 | 0.0 | 128 | 2e-05 |
| | | ✓ | 128 | 256 | 0.0 | 2 | 5e-06 |
| Gemma-7B | LoRA | ✗ | 128 | 128 | 0.0 | 128 | 2e-05 |
| | | ✓ | 256 | 256 | 0.25 | 32 | 2e-05 |
| Qwen2.5-3B | LoRA | ✓ | 128 | 64 | 0.1 | 128 | 5e-06 |
| Qwen2.5-7B | LoRA | ✓ | 8 | 4 | 0.0 | 64 | 2e-05 |
| Qwen2.5-14B | LoRA | ✓ | 128 | 128 | 0.15 | 64 | 5e-06 |

by executing our LoRA tuning Python script. All experiments are conducted on two A100-80GB GPUs.

**Discovered hyperparameters for each experiment**. The hyperparameters discovered after optimization and used for training are reported in Tables A1 and A2. Tables A3 and A4 present the details of the hyperparameters identified by the competing search methods, while Table A5 reports those obtained during the ablation studies. Our experiments show that, when applied to diverse models and LoRA variants, our framework consistently discovers hyperparameter configurations with a higher rank than the baselines. This suggests that our method effectively identifies hyperparameters most appropriate for each model and each LoRA variant.

Table A3: **Discovered hyperparameters by competing methods for math reasoning tasks.** We present the hyperparameter configuration, obtained through the competing method, that is used to train MetaMathQA for evaluating on the GSM8K and MATH datasets.

| Search Method | Hyperparameter | | | | |
|---|---|---|---|---|---|
| | Rank($r$) | Scaling Factor($\alpha$) | Dropout | Batch Size | Learning Rate |
| Random | 128 | 1024 | 0.1 | 16 | 5e-05 |
| Optuna (TPE) | 256 | 128 | 0.0 | 32 | 5e-04 |
| BO | 16 | 64 | 0.25 | 2 | 1e-04 |
| LBO | 128 | 4096 | 0.0 | 2 | 5e-06 |
| Tribes et al. (2024) | 8 | 256 | 0.1 | 4 | 1e-04 |

Table A4: **Discovered hyperparameters by competing methods for code generation tasks.** We present the hyperparameter configuration, obtained through the competing method, that is used to train CodeFeedback for evaluating on the HumanEval and MBPP datasets.

| Search Method | Hyperparameter | | | | |
|---|---|---|---|---|---|
| | Rank($r$) | Scaling Factor($\alpha$) | Dropout | Batch Size | Learning Rate |
| Random | 4 | 8 | 0.0 | 16 | 5e-05 |
| Optuna (TPE) | 256 | 128 | 0.2 | 16 | 1e-04 |
| BO | 16 | 8 | 0.15 | 2 | 5e-06 |
| LBO | 256 | 128 | 0.3 | 256 | 6e-04 |
| Tribes et al. (2024) | 4 | 64 | 0.0 | 4 | 3e-05 |

Table A5: **Discovered hyperparameters in ablation studies.** We present the hyperparameter configuration during our ablation studies, used in Table 6.

| Projection Layer | Domain-aware Prompting | Learnable Token | Hyperparameter | | | | |
|---|---|---|---|---|---|---|---|
| | | | Rank($r$) | Scaling Factor($\alpha$) | Dropout | Batch Size | Learning Rate |
| ✗ | ✗ | ✗ | 64 | 32 | 0.25 | 2 | 4e-04 |
| ✓ | ✗ | ✗ | 8 | 8 | 0.1 | 8 | 1e-04 |
| ✓ | ✓ | ✗ | 128 | 256 | 0.1 | 32 | 3e-04 |
| ✓ | ✓ | ✓ | 256 | 8192 | 0.0 | 4 | 5e-06 |

Table A6: **Performance across different model sizes.** Adapting our framework to different model sizes consistently shows improvements, indicating its effectiveness.

| Models | Ours | Accuracy (%) | | Pass@1 | |
|---|---|---|---|---|---|
| | | GSM8K | MATH | HumanEval | MBPP |
| LLaMA2-7B | ✗ | 41.47 | 5.24 | 16.31 | 35.47 |
| | ✓ | 62.93 | 12.88 | 30.49 | 42.59 |
| LLaMA2-13B | ✗ | 55.34 | 8.68 | 29.88 | 46.56 |
| | ✓ | 64.44 | 14.68 | 42.07 | 53.17 |

Table A7: **Performance differences across model sizes.** We apply the hyperparameters discovered for each model size of LLaMA2 to fine-tune all model sizes. "Model" denotes the model being fine-tuned, while "Settings" indicate the size of the model from which the hyperparameters were obtained.

| Model | Settings | Accuracy (%) | | Pass@1 | |
|---|---|---|---|---|---|
| | | GSM8K | MATH | HumanEval | MBPP |
| LLaMA2-7B | 7B | 62.93 | 12.88 | 30.49 | 42.59 |
| | 13B | 60.12 | 10.74 | 34.15 | 44.97 |
| LLaMA2-13B | 7B | 66.57 | 15.24 | 42.68 | 51.59 |
| | 13B | 64.44 | 14.68 | 42.07 | 53.17 |

Table A8: **Results of cross-applying hyperparameters across models.** We observe performance degradation when hyperparameters discovered for one model series are applied to another. This indicates that our framework effectively searches for hyperparameters suited to each model.

| Model | Settings | GSM8K | MATH |
|---|---|---|---|
| LLaMA2-7B | LLaMA2-7B | 62.93 | 12.88 |
| | Qwen2.5-3B | 39.5 | 5.2 |
| | Qwen2.5-7B | 32.68 | 4.7 |
| | Qwen2.5-14B | 52.46 | 8.06 |
| Qwen2.5-7B | Qwen2.5-7B | 83.93 | 48.08 |
| | LLaMA2-7B | 81.12 | 41.06 |
| | LLaMA2-13B | 80.06 | 40.18 |

Table A9: **Correlation between performance on a subset and the full dataset.** The percentages indicate the sampling ratios from the full dataset. For TSDS (Liu et al., 2024d), we report results separately when the target distribution is matched to the test dataset or the training dataset. Pearson correlation is used as the evaluation metric.

| Sampling Method | Math Reasoning | | Code Generation | |
|---|---|---|---|---|
| | GSM8K | MATH | HumanEval | MBPP |
| Random (1%) | 0.6879 | 0.4335 | 0.8052 | 0.5469 |
| Random (5%) | 0.8197 | 0.6483 | 0.8857 | 0.8879 |
| Random (10%) | 0.8566 | 0.6578 | 0.8652 | 0.9286 |
| TSDS (10%) by Test dataset | 0.8651 | 0.7117 | 0.8589 | 0.8209 |
| TSDS (10%) by Train dataset | 0.8529 | 0.6602 | 0.8624 | 0.9245 |

## C ADDITIONAL RESULTS

We provide supplementary experiments and analyses in addition to the main results presented in the main paper.

**Validation on models of different sizes**. Table A6 summarizes the validity of our framework under model size variations in LLaMA2. Even when the model size increases from 7B to 13B, our method successfully identifies appropriate hyperparameters, demonstrating the framework's robustness to changes in scale.

**Cross-application of hyperparameters within the same model series**. We apply the same procedure as in Table 8 to the LLaMA2 series, transferring hyperparameter settings discovered for one model size to another. The results in Table A7 show that hyperparameter configurations can remain effective across different scales within the same series. This further suggests the possibility of searching for hyperparameters on smaller models and transferring them to larger ones.

**Cross-application of hyperparameters between different model series**. To examine whether hyperparameters identified in one model transfer to another, we conduct experiments applying configurations discovered on Qwen2.5 to LLaMA-2, and vice versa, as shown in Table A8. Applying hyperparameters found on Qwen2.5 to LLaMA-2 leads to substantial performance degradation. Similarly, applying those from LLaMA-2 to Qwen2.5 also degrades performance. These results support the claim in Sec. 4.2 that preferred hyperparameter settings vary with model architecture.

**Correlation between subset training and full training**. Table A9 shows that correlation between subset training and full training remains consistent across all benchmarks. Notably, randomly sampling only 10% of the data still yields high correlation with full-dataset performance. This supports the claim in Sec. 4.2 that random sampling is a reasonable and efficient choice, comparable to more sophisticated data selection methods (Liu et al., 2024d). Thus, instead of tuning on the full dataset, leveraging proxy training evaluation provides a reliable proxy for estimating model performance during hyperparameter search.

Table A10: **Prompt templates with and without domain-aware prompting.**

```
rank(r)={rank_value}, Scaling factor(α)={alpha_value}, Dropout
Rate={dropout_value}, Batch Size={batchsize_value}, Learning
Rate={lr_value}
```

(a) Prompt templates without domain-aware prompting

```
* Rank (r): Controls adapter capacity by setting the low-rank
dimension, higher r increases expressivity (and memory/compute)
but raises overfitting risk. If you raise r, consider stronger
regularization or a lower learning rate.
* Scaling factor (α): Scales the LoRA update; the effective
update magnitude is **alpha / r**, so setting alpha ≈ r keeps
update strength stable. Larger alpha amplifies adaptation but can
destabilize training if LR is high.
* Dropout: Probability of dropping the adapter path to regularize
training; higher dropout curbs overfitting, especially with large r
or small datasets. With higher dropout you can often afford slightly
larger alpha or LR without instability.
* Batch size: Number of samples per optimizer step|larger batches
give smoother gradients and typically permit a proportionally larger
learning rate (linear-scaling rule) at the cost of more memory.
Small batches may need gradient accumulation or a reduced LR.
* Learning rate: Step size for adapter parameters|too high can
diverge (especially with large alpha/r), too low slows convergence.
Tune in conjunction with batch size and consider schedules (e.g.,
cosine) to balance speed and stability.
* rank(r): {rank_value}
* Scaling factor(α): {alpha_value}
* Dropout Rate: {dropout_value}
* Batch Size: {batchsize_value}
* Learning Rate: {lr_value}
```

(b) Prompt templates with domain-aware prompting

## D    TEMPLATE FOR DOMAIN-AWARE PROMPTING

We provide an example of the template for domain-aware prompting in Table A10. This template focuses on the roles and relationships of each hyperparameter and describes how training dynamics change as their values vary, based on prior studies (Kalajdzievski, 2023; Sun et al., 2024; Diehl, 2024; Meng et al., 2024; unsloth, 2025; Liu et al., 2025). Compared to a template without domain-aware prompting, this design captures rich domain knowledge about LoRA hyperparameters, significantly improving the effectiveness of the following Bayesian Optimization. The template can also be modified by users if needed.

