# OpenReview forum: "Efficient Hyper-Parameter Search for LoRA via Language-aided Bayesian Optimization"
_ICLR.cc/2026/Conference — ICLR 2026 Conference Withdrawn Submission_

### Official Review · Reviewer_xDxy · 2025-10-25

**Soundness:** 3
**Presentation:** 3
**Contribution:** 2
**Rating:** 4
**Confidence:** 4

**Summary:**

This paper introduces a novel framework for efficiently optimizing LoRA hyperparameters by integrating LLMs with Bayesian Optimization. Addressing the challenge that LoRA's performance is highly sensitive to hyperparameter choices and that exhaustive search is computationally prohibitive , this paper proposes using an LLM to map the discrete hyperparameter space to a continuous vector space. This mapping is guided by domain-aware prompting, which explicitly injects expert knowledge about hyperparameter relationships into the LLM in natural language. The framework also includes a learnable token and a projection layer to capture residual domain knowledge not easily described in prompts. To further accelerate the process, the method employs a "proxy training" strategy, evaluating configurations on a small data subset which strongly correlates with full-dataset performance.

**Strengths:**

- Significant Efficiency: The framework introduces a proxy training evaluation strategy, which drastically reduces the computational cost of HPO. By training on a small subset of data that shows a strong correlation with full-dataset performance, the method can reduce the overall time cost, enabling more efficient optimization.
- Strong Empirical Performance: The proposed method demonstrates superior effectiveness, finding high-performing hyperparameters within a small budget. This optimized configuration achieved performance improvement compared to standard hyperparameters. It consistently outperformed other HPO methods like random search, Optuna, and standard BO under the same budget and was both faster and more effective than other dedicated LoRA tuning algorithms.
- High Generalizability: The framework is designed as a plug-and-play module that shows broad applicability. It generalizes beyond standard LoRA to its variants, such as DoRA, rsLoRA, and PiSSA , and achieves consistent performance improvements across diverse model architectures ($e.g.$, LLaMA2, Mistral-7B, Gemma-7B) and scales.

**Weaknesses:**

- `Limited Innovation`: The paper appears to be a straightforward application of LLM+BO at the LoRA level, differentiated only by learnable tokens. This limits the novelty, and the work lacks a strong motivation for targeting LoRA's specific parameters.

- `Limited Comparison`: The experiments utilize a restricted set of LoRA variants, failing to comprehensively demonstrate the proposed method's effectiveness. Analysis using a wider range of LoRA variants [1-3] is warranted. Furthermore, the paper should include more comparisons with other LLM-aided BO methods.

- `Limited Generalizability`: The proposed framework is considerably more complex than traditional HPO methods. It integrates multiple components that must work in concert, each introducing its own set of parameters. This design necessitates joint optimization by maximizing the marginal log-likelihood, which substantially complicates both implementation and debugging.

- `Limited Open-Sourcing`: As this is a research-oriented work, the availability of code for further review and scrutiny is highly anticipated.


[1] When MOE Meets LLMs: Parameter Efficient Fine-tuning for Multi-task Medical Applications

[2] HydraLoRA: An Asymmetric LoRA Architecture for Efficient Fine-Tuning

[3] CoLA: Collaborative Low-Rank Adaptation

**Questions:**

See Weaknesses.

---

### Official Review · Reviewer_4d3b · 2025-10-26

**Soundness:** 2
**Presentation:** 1
**Contribution:** 2
**Rating:** 2
**Confidence:** 4

**Summary:**

This paper tackles hyperparameter optimization (HPO) for Low-Rank Adaptation (LoRA), a key method for efficient LLM fine-tuning. LoRA’s performance depends heavily on hyperparameters like rank and learning rate, but exhaustive search is costly. To address this, the authors introduce a framework combining Bayesian Optimization with LLMs, using the LLM to embed discrete hyperparameter configurations into continuous, informative representations.

**Strengths:**

1. The authors propose a novel framework that integrates Bayesian Optimization (BO) with Large Language Models (LLMs). The core innovation is using an LLM to convert discrete hyperparameter configurations into continuous, knowledge-rich embeddings.

**Weaknesses:**

1. The proposed method is only evaluated on domain-specific fine-tuning, which restricts its practical applicability. In particular, the proxy training approach appears difficult to extend to supervised fine-tuning (SFT), as instruction tuning typically requires diverse tasks and heterogeneous samples.

2. The baseline results reported in this work are significantly weaker than those in the original PiSSA paper [1]. For example, on Gemma, PiSSA reports (77.78, 31.33, 54.31, 66.17), whereas this work reports (75.51, 29.44, 49.39, 63.23), indicating a substantial performance gap.

3. The intent behind Table 8 is unclear. After applying hyperparameters optimized for different model sizes, there is no discernible trend or improvement. The observed performance differences are marginal and do not support strong conclusions.

4. In Table A1, the hyperparameter configurations for Qwen 2.5 3B and 14B are identical. Therefore, switching configurations between these models should not affect the results. However, the reported outcomes in Table 8 differ—could you clarify why the same settings yield different performance across model sizes.

**Questions:**

1. Could you clarify the performance and efficiency outcomes of full-parameter (FT) fine-tuning? Specifically, for Qwen 2.5 (3B or 7B), does your proposed method outperform full fine-tuning in terms of both metrics?


2. There is a significant discrepancy between your reported PiSSA results and those in the original PiSSA paper [1]. For example, on Gemma, the original paper reports (77.78, 31.33, 54.31, 66.17), whereas your results are (75.51, 29.44, 49.39, 63.23). Could you explain the source of this gap? The original authors provided standard deviations—could you share those for your results as well? Also, what is your model’s performance on MT-Bench?

3. The HBO method [2] was performed with SFT on general instruction data rather than domain-specific corpora. On what basis do you conclude that your method is more effective? Would you consider running SFT on the same dataset and comparing performance under zero-shot and few-shot settings on the same benchmarks?


[1] PiSSA: Principal Singular Values and Singular Vectors Adaptation of Large Language Models, Meng et al.

[2] Hyperparameter Optimization for Large Language Model Instruction-Tuning, Tribes et al.

---

### Official Review · Reviewer_a7hW · 2025-10-31

**Soundness:** 3
**Presentation:** 3
**Contribution:** 1
**Rating:** 2
**Confidence:** 3

**Summary:**

This paper proposes an efficient LLM-augmented Bayeisan optimization method for tuning the hyperparameters of LoRA adapters during LLM fine-tuning, with a view to improve the performance of the fine-tuned models as a result. The use of LLMs in the BO step enables the authors to both deal with the discrete-to-continuous mapping required by the kernels in Gaussian process-based surrogate functions, as well as to provide auxiliary domain-specific information about the hyperparameters being optimized in-context; whether through language, or via a special learned token. While proposed procedure is specialized to optimizing the hyperparameters of LoRA fine-tuning, the authors do demonstrate it working on different LoRA variants. The work demonstrates good improvements in absolute terms over the recommended 'default' hyperparameters across a range of LoRA variants and common datasets.

**Strengths:**

The paper is for the most part clearly written and well laid out. The preliminary section on Bayesian optimization is clear and appropriately brief for a background section, although it may have been nice to dwell on the acquisition function slightly more (explaining both its form and what maximizing it corresponds to) before moving on to the proposed framework.

The method is sound, and combines established techniques from prior work in Bayesian optimization and LLM-based deep kernel learning. The results are clearly presented in the tables and adequately characterize the performance of the method.

**Weaknesses:**

The work unfortunately has several weaknesses: spanning novelty, impact and timeliness.

On the novelty of the proposed method, it it not made very clear where the contributions of prior work such as Ranković & Schwaller 2025 end, and where this paper's contributions start. It appears that the prompting template, learned token, and projection are the new contributions from the authors. The remainder of the method is composed from classical results and methods from Bayesian optimization and recent work on LLM-assisted BO. Given the closeness of the method to prior work, unless there is a good reason why it could not have been included as a baseline in the experiments, it would have been nice to compare the performance of this paper's approach to it.

While the idea of using LLM embeddings to solve the long-standing problem of how to represent discrete parameters for use in kernels, and providing semantic descriptions of the parameters to draw on the rich semantic priors of the LLM is very compelling, it is almost a shame to limit the investigation to merely optimizing LoRA hyperparameters. Indeed, related work such as [Nguyen et al, 2025](https://arxiv.org/pdf/2410.10190) study a much broader range of tasks, and I feel as though the impact of the paper would have been much higher had it not limited itself to the admittedly interesting but very limited domain of LoRA hyperparameters for LLM fine-tuning. This concern over the impact of the paper holds when considering the strength of the results: while a healthy improvement is observed in absolute terms over the default hyperparameter settings, the improvement over baseline methods - including random search - in Table 4 perhaps do not justify the additional complexity of this method, especially if baselines such as the method of Ranković & Schwaller 2025 are applicable and are indeed omitted. One of the claims of this paper is the efficiency and reduced computational cost of this approach, however the evaluation of this claim is limited to Table 5 and a measurement of wall-clock time against a single baseline. A more rigorous evaluation would have compared FLOPs and/or memory to the final performance obtained, perhaps averaged over multiple seeds to account for the variance of each optimization. The pareto frontier would likely be bounded on one end by random search with a very low computational cost and relatively high performance, and perhaps a strong BO baseline elsewhere. This would be useful to characterise the benefit afforded by the special learned token and learned projection over their additional cost.

Finally on the timeliness of this approach and method, I think it is worth reflecting on whether further pursuing GP-based Bayesian optimization of model hyper-parameters is still the right direction in 2025. A pragmatic researcher or practitioner is perhaps more likely to reach for the simplicity of an LLM agent based approach than a complicated GP-based approach. That is, running a frontier LLM in a loop first with a description of the model at hand and all its various hyperparameters, any past hyperparameter configurations and their performance ($\mathcal{D} = \{(x_{i}, y_{i})\}_{i=1}^{n}$), performing some CoT reasoning over the next best configuration to try (potentially in parallel), and then outputting the next parameter settings to try from the candidate pool. I suggest that this will be just as cheap if not cheaper than the GP based approach, requires only inference and no gradient steps at runtime, is generalizable to many optimization tasks, and fully leverages the world-knowledge and "reasoning" abilities of modern LLMs in a way that may out-perform the simple heuristics employed by the acquisition function. I may of course be wrong in this assumption, however I think it is an important baseline to include to justify the complexity of the paper's proposed method.

Nits:
- L021: we provide a domain-aware textual prompts
- L024: we model the residual information hard to be linguistically
- L036: try to fix the bibtext to not read "Team" in the citation
- L182: the surrogate model parameterised $\omega$

**Questions:**

- What explains the occasional slight *drop* in task performance for settings models of increasing size in Table 8? Is this related to the difficulty of learning the projection or special token for larger models?

- Could you further clarify novel contributions introduced in your method as it relates to Ranković & Schwaller 2025; is it indeed the learned special token, projection and domain-specific prompt?

---

### Official Review · Reviewer_st7a · 2025-11-03

**Soundness:** 2
**Presentation:** 3
**Contribution:** 2
**Rating:** 4
**Confidence:** 3

**Summary:**

To leverage domain knowledge for LoRA HPO, the paper proposes a BO framework that converts hyperparameters into a domain‑aware textual template, feeds it (together with a learnable token) to a frozen LLM, projects the resulting embedding, and runs GP‑based BO in the continuous space. It is found that a 10% data subset serves as a useful proxy for full‑data training, further improving the efficiency. Across LoRA variants and multiple backbones, the method finds good configurations within ~30 iterations that significantly improve accuracy over standard settings and outperforming Random/Optuna/BO/LBO and the LoRA‑specific NOMAD baseline.

**Strengths:**

* The proposed framework leverages prior knowledge about LoRA via straightforward domain-aware prompting and effectively finds better hyperparameter combinations than default settings
* The method is computationally efficient, as 30 iterations are typically enough to yield a good hyperparameter configuration. The proxy evaluation further reduces the evaluation cost by roughly 10 times.
* Gains in performance are evaluated across multiple LoRA variants and backbones and seem to be consistent.

**Weaknesses:**

* The contribution seems to be incremental, and novelty of the proposed method is limited. LLMs have been already used to propose candidate hyperparameter configuration for HPO with BO, and LoRA tuning can be viewed as a downstream application.
* One can argue that using LLMs in HPO is non-trivial for LoRA. Still, the discrete/integer nature of the rank, which motivates the proposed method, might not actually be a fundamental obstacle: frameworks like Optuna already support integer search spaces.
* Some baseline results (e.g., TPE and BO outperformed by random searching) raise concerns about the strength of the baseline (see questions).

**Questions:**

* In Table 4, TPE and BO are outperformed by random searching, which is surprising. What settings were used for TPE and BO, and why might these methods fail here? Specifically, the appendix mentions that TPE with categorical hyperparameter candidates is used. TPE in Optuna supports integer search spaces, so why not model rank as an integer with appropriate priors?
* Is there any pre-processing steps before feeding the 5-dimensional vector to BO?

---

### Note · Authors · 2025-11-13

I have read and agree with the venue's withdrawal policy on behalf of myself and my co-authors.